# The mediation of perceived risk's impact on destination image and travel intention: An empirical study of Chengdu, China during COVID-19

**Xiufang Jiang** [1]*, **Jianxiong Qin** [2]*, **Jianguo Gao** [3], **Mollie G. Gossage** [4]

**1** College of Economics, Southwest Minzu University, Chengdu, China, **2** College of Historical Culture & Tourism, Southwest Minzu University, Chengdu, China, **3** College of Earth Sciences, Chengdu University of Technology, Chengdu, China, **4** Department of Anthropology, University of Wisconsin-Madison, Madison, Wisconsin, United States of America

* 15203656667@163.com (XJ); jx-qin@vip.163.com (JQ)

**Data Availability Statement:** All relevant data are within the paper and its Supporting Information files.

**Funding:** This research was funded by Research project of the Key Research Base of Humanities

## Abstract

Perceived risk clearly impacts travel behavior, including destination selection and satisfaction, but it is unclear how or why its effect is only significant in certain cases. This is because existing studies have undervalued the mediating factors of risk aversion, government initiatives, and media influence as well as the multiple forms or dimensions of risk that can mask its direct effect. This study constructs a structural equation model of perceived risk's impact on destination image and travel intention for a more nuanced model of the perceived risk mechanism in tourism, based on 413 e-questionnaires regarding travel to Chengdu, China during the COVID-19 pandemic, using the Bootstrap method to analyze suppressing effect. It finds that while perceived risk has a significant negative impact on destination image and travel intention, this is complexly mediated so as to appear insignificant. Furthermore, different mediating factors and dimensions of perceived risk operate differently according to their varied combinations in actual circumstances. This study is significant because it provides a theoretical interpretation of tourism risk, elucidates the mechanisms or paths by which perceived risk affects travel intention, and expands a framework for research on destination image and travel intention into the realms of psychology, political, and communication science. It additionally encourages people to pay greater attention to the negative impact of crises and focuses on the important role of internal and external responses in crisis management, which can help improve the effectiveness of crisis management and promote the sustainable development of the tourism industry.

## Introduction

Safety is a fundamental prerequisite for development in the tourism industry and a crucial consideration for the majority of tourists and the tourism sector as a whole [1]. Its assurance is regarded as one of the most pressing issues now facing the industry around the world [2].

and Social Sciences in Universities of Sichuan Province, grant number GJGY2019-ZD002(Q received the award); and was supported by "the Innovative research project of Southwest Minzu University", grant number CX2020BS20(J received the award). The funders had no role in study design, data collection and analysis, decision to publish, or preparation of the manuscript.

**Competing interests:** The funders had no role in study design, data collection and analysis, decision to publish, or preparation of the manuscript.

However, "a tourism crisis can take an infinite variety of forms and have been occurring regularly for many years" [3], according to the United Nations World Tourism Organization (UNWTO). Historically, terrorism [4], political instability [5], economic crises [6], financial problems [7], natural disasters [8], and infectious diseases [9] have caused major blows to the global tourism industry. As the UNWTO warns: "Never underestimate the possible harm a crisis can do to your tourism. . . They are extremely dangerous. . . the damage wrought by a crisis can stay in the minds of potential tourists for a long time" even after the event itself [3]. Indeed, potential tourists' perceived tourism risks have been shown to increase significantly over the short term following crisis events. Existing theories of risk suggest that a relatively high level of perceived risk will compel people to cancel travel plans or change their destinations in order to minimize or transfer that risk to the greatest possible extent [10, 11]. This held true in 2020, when, according to the latest data, the UNWTO [12] states that global tourism suffered its worst year on record, with international arrivals dropping by 74%. Destinations worldwide welcomed one billion fewer international arrivals in 2020 than in the previous year due to both widespread travel restrictions and an unprecedented fall in demand. This dwarfs the mere 4% decline recorded during the 2009 global economic crisis.

Intuitively, perceived risk should exert a straightforward negative influence on willingness to travel and travel behavior, but this is by no means always the case. Following the 2012 "Sanya incident" and 2015 "Qingdao incident" (both involving seafood restaurants in China exposed for charging tourists with exorbitant prices), the respective destination images were hardly damaged by online commenters' attacks; tourist visits to Sanya and Qingdao not only failed to drop, but demonstrated significant increases in the wake of the scandals. (From 2012 to 2014, Sanya's annual domestic overnight stays increased by 30.59%, 11.97%, and 11.33% [13], respectively; from 2015 to 2018, Qingdao's annual domestic overnight stays increased by 9.03%, 8.44%, 9.2% and 13.7% [14], respectively.) Empirical research has also found the negative impact of perceived risk on behavioral intention to be insignificant [15, 16], or even found perceived risk to have a significant positive effect [17].

Such phenomena cannot help but make one wonder: Does perceived risk actually have a negative impact on travel intention? Could it even have a positive impact? How do studies on perceived risk come to such diverse conclusions? What forces are truly at work here? This article finds that there are two main reasons underlying the inconsistencies. First, there exist certain key mediating variables that have an obvious influence on destination image and travel intention. And, because these mediating variables' effect is similar in magnitude to the direct effect, but their direction is opposite, the negative impact of perceived risk may be completely masked. Second, perceived risk is not a monolith. Empirical research has indicated that different types of risk have different effects on behavioral intention. In combination, these types can effectively "neutralize" the direct effect of perceived risk, weakening or fully mitigating the direct effect. This study intervenes with two improvements: one, introducing three key crisis response factors—risk aversion, government measures, and media influence; two, breaking perceived risk down into six dimensions—physical risk, equipment risk, cost risk, social risk, performance risk, and psychological risk—to conduct a more detailed comparison of varied paths of action under the influence of risk.

## Background and research hypotheses

### Event background

Chengdu (30˚5'–31˚26'N and 102˚54'–104˚53'E) is the capital of Sichuan Province in Southwest China (Fig 1). It is the region's largest transportation hub and has been hailed "the best

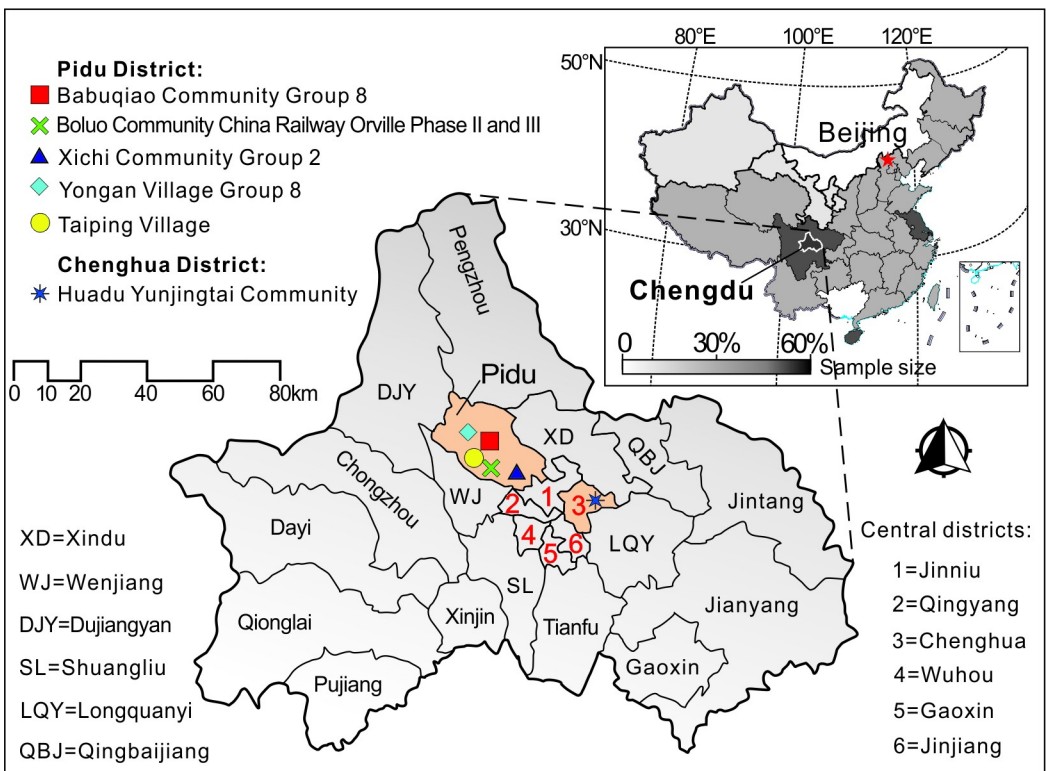

**Fig 1. Study site.**

tourist city in China," renowned both domestically and internationally for its rich historical, cultural, and natural resources.

When Wuhan entered a state of lockdown on January 23, 2020, other cities and communities across China immediately responded with their own epidemic control policies, including restrictions on non-essential travel and business operations. But two months later, apart from increased temperature checkpoints and mask requirements, life for Chinese citizens had largely returned to normal. From March through November, Chengdu had not had a single new locally transmitted case of COVID-19. But then, on December 7, the virus re-emerged. It started with a single new case of locally transmitted COVID-19 identified in Chengdu's Pidu District; before long, six different areas across the city had been upgraded to medium-risk: in Pidu District, Babuqiao Community Group 8, Xichi Community Group 2, Taiping Village, Yongan Village Group 8, Boluo Community China Railway Orville Phase II and III, and—in Chenghua District—Huadu Yunjingtai Community (Fig 1). The Chenghua District case was a young woman (surname "Sun") whose movements around the city from the time of infection to diagnosis were widely shared on social media, stoking fears of yet-undetected spread. One commenter joked: "Last week, twenty-year-old Ding Zhen [a breakout internet celebrity from western Sichuan's Litang] singlehandedly made Sichuan the hottest destination in the country. This week, twenty-year-old Miss Sun singlehandedly caused the country to cancel their airline tickets to Sichuan." In order to keep the outbreak under control, the Chengdu government quickly took firm, decisive and strict measures that effectively blocked the further spread of COVID-19. By December 12, nearly 2.3 million people across Chengdu had been given nucleic acid tests, including all residents of the Pidu District. The outbreak never expanded as some watchers online had feared; on December 31, Chengdu was cleared of all medium-risk areas.

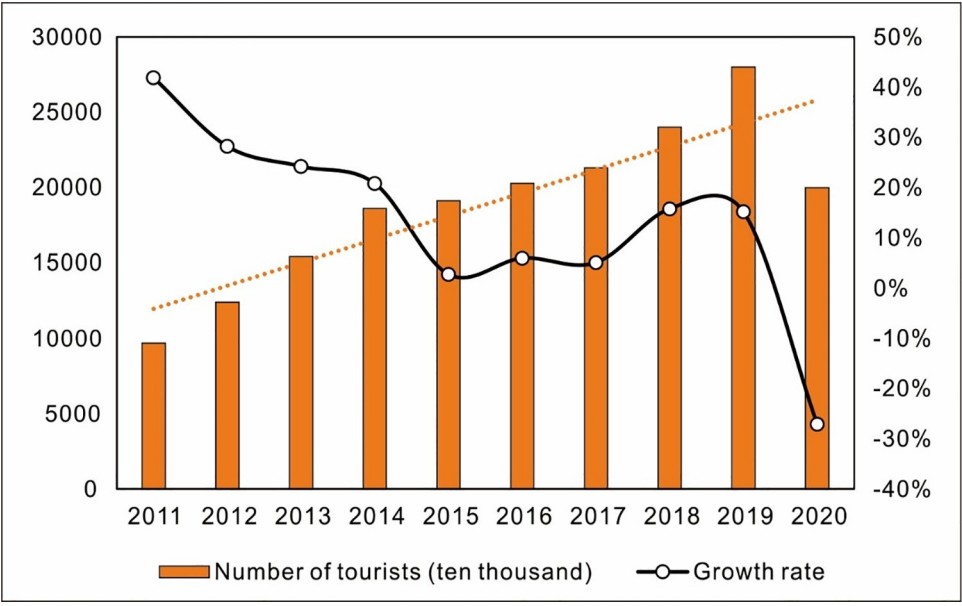

**Fig 2. Annual tourist volume for Chengdu over the past ten years.**

Additionally, we have annual tourist volume and growth rate for Chengdu over the past years, and have mapped the trend of change (Fig 2), which clearly revealed the impact of the outbreak on Chengdu' tourism.

Once the outbreak was effectively contained, Chengdu acted to promote the sustainable development of the tourism industry and maintain the confidence of tourists and tourism operators alike. It issued the "Measures in Response to the Impact of the Novel Coronavirus Outbreak and for the Healthy Development of the Culture and Tourism Industry" and enacted a number of crisis management measures targeting the tourism industry—including the issuance of consumer vouchers and implementation of precision marketing (to reduce the epidemic's impact on cultural and tourism enterprises), tourist volume restrictions for scenic sites, and mandatory or voluntary inspections at public gathering places. These measures significantly improved tourists' perceptions of safety and thus improved their travel intention.

## Theoretical background and research hypotheses

**How perceived risk relates to destination image and travel intention.** Destination image is a core concept for the branding of tourism sites [18] and a major factor that shapes tourists' perceptions of a place, their travel planning, destination selection, and overall trip satisfaction [19–21]. For places with unique features and relatively strong PR, their image is a crucial component of destination branding [18], whereas for places with a poor destination image, it constitutes the major obstacle to attracting tourists [22, 23]. Destination image, it should be noted, is a multi-dimensional concept. According to its underlying source of information, Gunn [24] classifies destination images into organic (non-commercially formed) and induced (commercially formed). However, advances in networking and communication technologies mean that the boundary between commercial and non-commercial information is increasingly blurred, such that this typification is no longer adequate for our times. Gartner [25] breaks destination image down into three dimensions—cognitive image, affective image, and conative image; this model is further developed by Baloglu [26], who proposes a tripartite division of cognitive, affective, and overall image. Pan [27], meanwhile, classifies destination

images as either baseline images or enhanced images according to tourists' subjective willingness to understand information about the site. A review of the literature on this concept's evolution and distribution makes clear that destination image is a result of multiple factors, affected by both tourists personally as well as external sources of information.

In this study, perceived risk refers to tourists' subjective expectation of potential harms or losses to be suffered in the course of travel. Observable discrepancies in perceived risk are found to exist between different tourist demographic groups according to gender, age, educational level, marital status, and source location (i.e. place of origin) [27–31]. According to theories of perceived risk, consumers make behavioral decisions based on bounded rationality and in accordance with the satisfaction theory of motivation. Bauer [32] believes that in decision-making, consumers are not so much "maximizing utility" as minimizing the associated risks. For tourist consumers, the level of environmental risk perceived will affect destination image, behavioral intention, behavioral decision-making, and experience satisfaction [33–35]. For example, empirical research by Chew et al. [20] finds that after the Fukushima nuclear disaster in Japan, the social, psychological, and financial risks posed to tourists had a significant negative impact on their cognitive and affective images of Japan, indirectly affecting their willingness to revisit, while physical risks directly affected willingness to revisit. Neuburger et al. [36], meanwhile, identifies a high correlation between tourists' perceived risk and whether or not they canceled travel plans during the COVID-19 outbreak. The current study thus puts forward the following hypotheses:

- H1: Destination image has a positive impact on travel intention.

- H2: Perceived risk has a negative impact on destination image.

- H3: Perceived risk has a negative impact on travel intention.

Perceived risk, like destination image, is a multi-dimensional concept. Fischer et al. [37] develop a coding scheme for perceived risk, laying a foundation for subsequent quantitative research. Boksberger et al. [38] identify six dimensions of perceived risk—physical risk, performance risk, psychological risk, social risk, financial risk, and time risk. Qi et al. [39] construct a tripartite model of physical risk, performance risk, and psychological risk. Fuchs et al. [33], meanwhile, favor a classification of perceived risk as physical risk, performance risk, psychological risk, and financial risk. Mohseni et al. [40] divide perceived risk along five dimensions —physical risk, performance risk, financial risk, time risk, and equipment risk. The model put forward by Li et. al [41] specifically for the COVID-19 outbreak divides tourists' perceived risk into physical, psychological, social, performance, image, and time risks. Following a comprehensive review of the research on perceived risk, Zhu et al. [42] posit that performance risk, physical risk, psychological risk, and social risk are relatively important factors compared to financial risk, time risk, and equipment risk, while time and financial risk can moreover be combined into "cost risk". It can thus be seen that, for the COVID-19 pandemic period, use of physical risk, equipment risk, cost risk, social risk, performance risk, and psychological risk can measure tourists' perceived risks rather comprehensively. Considering how different types of perceived risk exert different effects on destination image and travel intention [22, 43], this study puts forward the following hypotheses:

- H2a: Physical risk, equipment risk, cost risk, social risk, performance risk and psychological risk have a negative impact on destination image.

- H3a: Physical risk, equipment risk, cost risk, social risk, performance risk and psychological risk have a negative impact on travel intention.

**The role of risk aversion in perceived risk's effect on destination image and travel intention.**   Regulatory focus theory purports that in uncertain environments, consumers will make rational decisions in order to avoid risky consumption behavior. Any perceived risk will increase a consumer's demand for risk aversion, along with their anticipation of regret [33]. Under real tourism circumstances, tourists generally exhibit risk-averse behavior to ensure the safety of individual tour activities [44]. In psychology, empirical research has identified a correlation between perceived risk and risk-averse attitudes or behavior; a relatively high perception of risk will lead people to adopt preventive measures—vaccination, for example [45, 46]. Bai [47] believes that destination image describes a cognitive feature found among all tourists regarding the destination in question, but that said cognition varies by individual travel experiences and values as well as the means and degree of outside stimulation. Zhu et al [42] find that knowledge has a direct impact on tourists' perception of risk and attitude toward risk aversion along with an indirect impact on intent to travel to or recommend rural tourism. Liang et al. [48] propose that travel insurance is an important measure for control of the COVID-19 epidemic in Chongqing as well as for lowering tourists' perception of such risk. Thus it can be seen that tourists' sensitivity to tourism risk varies according to individual differences as well as how the relevant information is processed. This in turn leads to various risk avoidance awareness and behavior [49, 50]. The result is diversified destination images, tourism attitudes, and tourism behaviors. Therefore, this study proposes the following hypotheses:

- H4: Through risk aversion, perceived risk has an indirect impact on destination image.

- H4a: Physical risk, equipment risk, cost risk, social risk, performance risk and psychological risk have, through risk aversion, an indirect impact on destination image.

- H5: Through risk aversion, perceived risk has an indirect impact on travel intention.

- H5a: Physical risk, equipment risk, cost risk, social risk, performance risk and psychological risk have, through risk aversion, an indirect impact on travel intention.

**The role of government initiatives in perceived risk's effect on destination image and travel intention.**   Government initiatives refers to all of the local (destination-based) government's efforts to respond to crises, restore destination image, and attract tourists. Fang [51] argues that in crisis management, government response is the external response with the greatest effect in terms of mitigating perceived risk. According to Zhao et al. [52], following the 2008 Wenchuan earthquake, Sichuan's tourism industry was able to recover and even surpass pre-earthquake levels within a two-year period—due in large part to the government's efforts. Thus they stress the advantageous position of the government to guide tourism crisis management. Chai et al. [30] conduct empirical research and draw the conclusion that 10 factors have a marked influence on tourists' risk perception including external assistance, objective reporting from the media, knowledge about the crisis, recovery plans and measures, influence from the potential tourist's social circles, gender, and the attitude and performance of government departments. Sharma et al. [53] propose a framework for the post-pandemic rejuvenation of the global tourism industry, for which they believe government measures are the primary factor contributing to the industry's resilience. Building upon these findings, this study considers that tourists' evaluations of a local government's crisis response—which correlate with the government's actual ability to restore destination image and attract incoming travelers—will affect tourists' perception of risks and general awareness of the destination, thus affecting their intent to travel. The following hypotheses are therefore proposed:

- H6: A positive evaluation of government initiatives helps reduce the impact of perceived risk on destination image.

- H6a: Physical risk, equipment risk, cost risk, social risk, performance risk and psychological risk have, through government initiatives, an indirect influence on destination image.

- H7: A positive evaluation of government initiatives helps reduce the impact of perceived risk on travel intention.

- H7a: Physical risk, equipment risk, cost risk, social risk, performance risk and psychological risk have, through government initiatives, an indirect influence on travel intention.

**The role of media influence in perceived risk's effect on destination image and travel intention.** Media influence is a collective term referring to all the sources of information that can change a tourist's impression of a destination. In the information age, the power that media has over the tourism industry is truly unprecedented—whether in terms of the breadth or depth of its reach, the frequency of its interventions, or the variety of its forms. Media has fundamentally changed, and continues to change, the way that tourists search for and consume information, and it plays an increasingly important role when it comes to destination image and travel planning [54]. Acquiring information is the key link in developing an impression of a destination, and media is the key means of disseminating destination-related information [55]. Media-disseminated information has been shown to have a strong influence on destination image [55, 56] and its total impact on travel intention to reach 19.9% [57].

In tourism and tourism crises, the media's power is mainly manifested in the following three ways: First is deliberate official promotion. Media has wide coverage, such that tourists can obtain information easily, and the cost is low [25]. Therefore, marketing agencies typically use media to promote a destination's original image in a way that forms an induced image and contributes to increases in tourist numbers and tourism revenue. Second is incidental social promotion. After tourists have had a firsthand experience of a place, their online comments and other social media activity will also have an effect on destination image and the travel intention of other potential tourists. Through an experiment using search engines and simulating the process of trip planning, Zheng at al. [58] verify the importance of social media's role in information gathering and tourism decision-making. Zeng et al. [59] find that social media plays a significant role in many aspects of tourism, especially in information searches and decision-making behavior, tourism promotion, and in focusing on best practices for interacting with consumers. Third is vital communication and education In the course of managing a crisis, the media can provide the public with information for selecting a course of action, enhance crisis awareness, issue public warnings, and provide crisis management personnel with information and specific action guidelines [60, 61]. Media response to crises can inspire the public to pay more attention to specific social risks, to develop a better understanding of risks, to interpret risk information in a calmer manner, and to participate in risk-related discussions, ultimately forming more reasonable perceptions and attitudes regarding risk [62]. All three of these media utilities do much to create a favorable destination image and stimulate travel intentions [36, 56]. This study holds that media guidance does affect tourists' perceived risk and thus, indirectly, destination image and travel intention. The following hypotheses are therefore proposed:

- H8: A positive evaluation of media influence helps reduce the impact of perceived risk on destination image.

- H8a: Physical risk, equipment risk, cost risk, social risk, performance risk and psychological risk have, through media influence, an indirect impact on destination image.

- H9: A positive evaluation of media influence helps reduce the impact of perceived risk on travel intention.

- H9a: Physical risk, equipment risk, cost risk, social risk, performance risk and psychological risk have, through media influence, an indirect impact on travel intention.

The above analysis shows that perceived risk impacts travel intention not only through risk aversion, government initiatives and media influence, but that it may also trigger a chain of multiple mediating effects through destination image, generating a multiple mediation research model. To this end, the following hypotheses are proposed:

- H10: Risk aversion, government initiatives, media influence, and destination image create a chain of mediating effects in the course of perceived risk's impact on travel intention.

- H10a: Physical risk, equipment risk, cost risk, social risk, performance risk, and psychological risk create a chain of mediating effects that impact travel intention through risk aversion, government initiatives, media influence, and destination image.

A conceptual model is provided below according to the above hypotheses (Fig 3).

## Materials and methods

### Questionnaire design and variable measurement

To ensure the reliability and validity of results, the questionnaire design draws upon well-established scales, to which it makes appropriate adjustments integrating the research goals and specific context. The main body of the questionnaire utilizes a five-point Likert scale (where 1 is "strongly disagree" and 5 is "strongly agree"). Identification of perceived risk draws on Zhu et al. [42], measuring 18 items across six dimensions—physical, equipment, cost, social, performance, and psychology (Table 1). Measurement of risk aversion draws on Chapman et al. [45] and Weinstein et al. [46], covering three aspects—information, insurance, and vaccination—in accordance with the actual circumstances of infectious disease. Government initiatives are assigned a value from

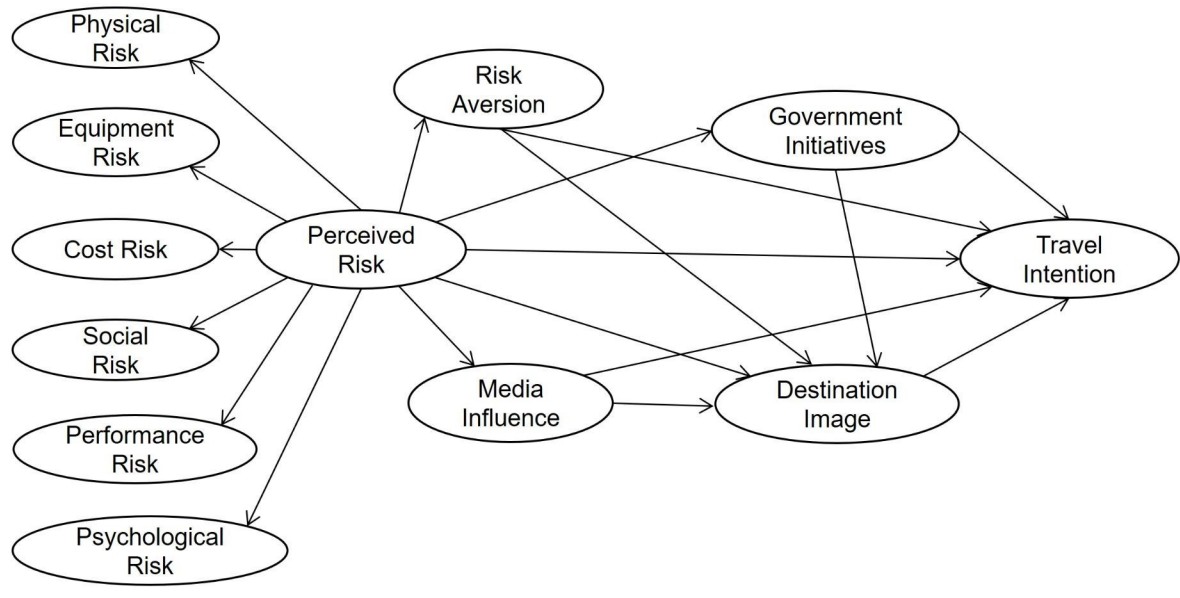

**Fig 3. Conceptual model.**

**Table 1. Confirmatory factor analysis.**

| Factor (Latent Variable) | Measurement Item (Manifest Variable) | Mean | Std.Estimate | AVE | CR |
|---|---|---|---|---|---|
| Physical Risk | PHY1 Human-made crises or natural disasters (earthquakes, mudslides, etc.) that may occur at tourism sites | 3.08 | 0.87 | 0.72 | 0.89 |
| | PHY2 Public security incidents that may occur at tourism sites | 3.18 | 0.87 | | |
| | PHY3 I may get sick during travel, e.g. with COVID-19 | 3.16 | 0.81 | | |
| Equipment Risk | EQU1 The destination has poor basic infrastructure | 3.17 | 0.85 | 0.79 | 0.92 |
| | EQU2 The destination has poor sanitation | 3.19 | 0.91 | | |
| | EQU3 Traffic is inconvenient at the tourism destination | 3.23 | 0.92 | | |
| Cost Risk | COS1 During the trip, actual costs will exceed expectations. | 3.41 | 0.77 | 0.74 | 0.90 |
| | COS2 Quarantine measures put in place for COVID-19 carry time-related costs | 3.50 | 0.89 | | |
| | COS3 Travel restrictions put in place for COVID-19 mean that certain experiences are off-limits | 3.52 | 0.93 | | |
| Social Risk | SOC1 If I travel to Chengdu during this period, others may think negatively of me | 2.90 | 0.94 | 0.74 | 0.89 |
| | SOC2 If I travel to Chengdu during this period, others will criticize me | 2.89 | 0.92 | | |
| | SOC3 If I travel to Chengdu during this period, friends and family members will not support my trip | 3.05 | 0.68 | | |
| Performance Risk | PER1 Tourism activities are unable to meet my requirements for relaxation | 2.93 | 0.71 | 0.74 | 0.89 |
| | PER2 The quality of tourism services does not meet expectations | 3.22 | 0.95 | | |
| | PER3 There are not as many tourism products as expected | 3.22 | 0.92 | | |
| Psychological Risk | PSY1 I feel worried traveling during the COVID-19 period | 2.61 | 0.88 | 0.88 | 0.96 |
| | PSY2 I feel anxiety traveling during the COVID-19 period | 2.52 | 0.97 | | |
| | PSY3 I feel nervous traveling during the COVID-19 period | 2.50 | 0.96 | | |
| Risk Aversion | AT1 Before traveling, gather more information about the destination | 3.38 | 0.87 | 0.68 | 0.87 |
| | AT2 Buy travel insurance | 3.17 | 0.83 | | |
| | AT3 Get a vaccine | 3.38 | 0.79 | | |
| Media Influence | MI1 I consider media opinions when selecting my vacation destination | 3.59 | 0.86 | 0.67 | 0.86 |
| | MI2 For planning a tour, I feel media is a very authentic source of information | 3.63 | 0.74 | | |
| | MI3 Developments reported by media can change my opinion about the destination | 3.67 | 0.84 | | |
| Government Initiatives | GOV1 Quality of infrastructure in Chengdu (public transport, roads, etc.) at the destination is satisfactory | 3.60 | 0.90 | 0.79 | 0.94 |
| | GOV2 I think the Chengdu government's policies/regulations are favorable for tourists | 3.56 | 0.91 | | |
| | GOV3 I think the Chengdu government is committed in promoting the destination's positive image | 3.42 | 0.86 | | |
| | GOV4 Services I received from Chengdu's public servants (including tourism police, etc.) were satisfactory | 3.59 | 0.88 | | |
| Destination Image | TDI1 I think Chengdu has a better image than other tourism destinations | 3.51 | 0.93 | 0.86 | 0.95 |
| | TDI2 I think the overall travel experience Chengdu provides is able to meet my needs | 3.51 | 0.93 | | |
| | TDI3 I would recommend Chengdu as a favorable destination | 3.45 | 0.92 | | |
| Travel Intention | TI1 I am interested in going to Chengdu for travel | 3.45 | 0.93 | 0.84 | 0.94 |
| | TI2 I will travel to Chengdu in the future | 3.41 | 0.93 | | |
| | TI3 There is a high probability that I will travel to Chengdu in the future | 3.46 | 0.90 | | |

GOV1 to GOV4 according to the well-established scales of Enright et al. [63], Chi et al. [64], and Merrilies et al. [65]. A value from MI1 to MI3, meanwhile, is calculated for media influence according to the scales of Chen et al. [66] and Jalilvand et al. [67]. Destination image is measured by three items in reference to the scale of Parrey et al. [68]. Finally, travel intention draws upon the scale from Lee et al. [69] with a value range from TI1 to TI4.

Demographic variables: gender, age, educational level, marital status, industry, occupation, and source location. Travel behavior variables: whether one has participated in tourism activities since the outbreak of COVID-19; travel experience (never traveled, traveled once, traveled multiple times).

### Pre-test and questionnaire revision

From December 7 to December 14, 2020, 100 electronic questionnaires were distributed as a pre-test. We found that "COS4 traveling during this period requires too much time on the road [due to epidemic control checkpoints, etc.]" was semantically similar to COS2, and that four items—"COS5 due to policies restricting travel, certain services cannot be provided as planned", "PER4 scenery and views of nature are not satisfactory", "SOC4 tourism activities cannot improve family connections", and "TI4 I would rather go to Chengdu for tourism than other destinations"—had standard load factor values of less than 0.4, indicating weak correspondence between the given factor and the item being measured. After removing these, the reliability coefficient increased significantly. Thus, the five test items mentioned above were deleted in the revision of the questionnaire.

### Survey sampling and data collection

In order to avoid the risks of direct contact, this study used an online questionnaire to obtain sample data. "Questionnaire Star" (https://www.wjx.cn/) is a relatively well-established survey website dedicated to the self-design of questionnaires and related services in China. It is equipped with a wealth of questionnaire-style templates which editors can modify and operate independently. It provides complete output functionality, delivering survey results in various formats such as Word, PDF, and Excel. As of October 2021, the website has 2.6 million registered users. A total of 136 million questionnaires have been published on the questionnaire star platform, and 10.778 billion answer sheets have been collected. On average, more than 1 million people fill out questionnaires on the questionnaire star platform every day. Its paying customers cover more than 30,000 companies and 90% of universities in China [70].

Because there is no clear or uniform requirement for sample size in statistics, the common standard is 5–10 times the scale items [71]. As our questionnaire contains a total of 34 scale items in our questionnaire, a suitable sample size would be between 170 and 340. Some scholars have also constructed a formula for calculating the total sample size. Wu believes that when the sample population is quite large or infinite, the sample size formula is as below (1).

$$n \geq \left(\frac{k}{\alpha}\right)^2 p(1-p) \tag{1}$$

where $n$ represents sample size. Usually $p$ and $\alpha$ is set respectively to 0.5 and 0.05, because the most reliable sample size can be obtained when 0.5 and 0.05 is set [72]. In this study, α was at 95% confidence level, that is, p = 0.5. And the confidence level used in interval estimation is 1–0.05 = 0.95. The time quantile is k = 1.96. Therefore, the formula yielded an n of 384.16 as the minimum acceptable sample size.

The official questionnaire was open from December 26, 2020 to January 26, 2021. After applying the "Questionnaire Recommendation Service" through Questionnaire Star, the online system randomly invited people from its 2.6-million-sample database to fill in their responses. Beyond that, the research team and members of the surrounding community were invited via weblink, QR code, or WeChat to fill out the survey. By January 24, 2021, however, four Chinese provinces or province-level regions remained unrepresented in the pool of questionnaire respondents—Qinghai, Gansu, Guangxi, and the Tibet Autonomous Region. The researchers then adopted respondent-driven sampling, specifically seeking respondents from the above-mentioned provinces and region, and required those surveyees to recommend a specific number of peers from among their demographic groups as a means of reducing sampling bias.

A total of 428 questionnaires were received, among which 15 were deemed invalid and eliminated from the study, while 413 were deemed valid—making for an efficiency rate of

96.5%. Among the valid surveys, 69% came from respondents identifying as male; 30% came from those identifying as female. In terms of the age structure, the main demographic were those aged 18 to 25 (67% of the total), followed by those aged 26–30 years old (18%), then 31–40 years old (10%). In terms of educational attainments, the largest group of respondents had a college or post-secondary professional degree (55%), followed by those holding a high school diploma (25%), and then those with a master's or doctoral degree (16%). All together, the respondents come from 91 cities in 32 of China's provinces or regions (distribution is shown in Fig 1) and were employed in 15 occupations across 24 industries. As for travel experience, 19% had participated in tourism activities since the initial outbreak of COVID-19 (in January 2020).

## Analysis methods

SPSS20.0 was used to conduct reliability and validity analysis; structural equation model, path analysis and Bootstraping were used to verify the study's hypotheses; t-testing and variance analysis were used to verify the distinctions in perceived risk between demographic groups and social statistics. Testing for mediation has traditionally utilized stepwise regression [73, 74]. Baron et al. [73] believe that mediation depends on a total significant effect, in other words, an independent variable significantly affecting the dependent variable. In this context, "mediation" expresses the process and mechanism of the independent variable's influence on the dependent variable. Some scholars have questioned this, arguing that mediation is not dependent on net significance, because net effect will become insignificant if the indirect effect is of a similar magnitude to the direct effect but opposite in sign [75, 76]. In other words, mediation may still be present when the net effect is insignificant. The literature often refers to such mediation as a "suppressing effect" to distinguish it from more commonly observed situations [77, 78]. Thus it can be seen that opposing direct and indirect effects (i.e. one positive and one negative) conceal one another, making the net effect's absolute value lower than expected. With this in mind, many scholars now prefer the Bootstrap method to stepwise regression for statistical sampling. This study therefore uses the Bootstrap method to analyze multiple mediating variables.

## Results

### Trust level analysis

Cronbach's alpha ($\alpha$) is used to measure the reliability of the data. Reliability analysis of 34 items in total under 11 factors—physical risk, equipment risk, cost risk, social risk, performance risk, psychological risk, risk aversion, government initiatives, media influence, destination image, and travel intention—results in a reliability coefficient of 0.93 for the questionnaire. A coefficient over 0.9 indicates that the research data is of high quality [79].

### Validity analysis

The effective sample size (413) exceeds 10 times the number of analysis items (34), which means confirmatory factor analysis (CFA) is suitable for testing validity [80, 81]. The results of CFA are shown in Table 1. The factor loading of each measurement item is greater than 0.6; p is lower than 0.001; the factors' corresponding average variance extracted (AVE) values are all greater than 0.6, such that composite reliability (CR) are all greater than 0.8; all of these indicate good correlation between factors and measurement items, indicating convergent validity better than the accepted standard.

The Fornell-Larcker method is used to test discriminant validity. It finds that latent variables' AVE square roots are all greater than the maximum absolute value of the correlation

**Table 2. Discriminant validity.**

|  | 1 | 2 | 3 | 4 | 5 | 6 | 7 | 8 | 9 | 10 | 11 |
|---|---|---|---|---|---|---|---|---|---|---|---|
| 1. Physical | **0.85** | | | | | | | | | | |
| 2. Equipment | 0.74 | **0.89** | | | | | | | | | |
| 3. Cost | 0.64 | 0.72 | **0.86** | | | | | | | | |
| 4. Social | 0.57 | 0.58 | 0.47 | **0.86** | | | | | | | |
| 5. Performance | 0.65 | 0.72 | 0.61 | 0.58 | **0.86** | | | | | | |
| 6. Psychological | 0.40 | 0.39 | 0.18 | 0.50 | 0.44 | **0.94** | | | | | |
| 7. Risk aversion | 0.24 | 0.16 | 0.27 | 0.10 | 0.10 | -0.02 | **0.83** | | | | |
| 8. Govt initiatives | 0.29 | 0.33 | 0.36 | 0.27 | 0.29 | 0.20 | 0.41 | **0.82** | | | |
| 9. Media influence | 0.06 | 0.06 | 0.15 | 0.09 | 0.03 | -0.04 | 0.40 | 0.49 | **0.89** | | |
| 10. Dest image | 0.01 | -0.03 | 0.07 | -0.01 | -0.07 | 0.00 | 0.51 | 0.28 | 0.39 | **0.93** | |
| 11. Travel intent | -0.01 | -0.08 | 0.07 | -0.07 | -0.10 | -0.05 | 0.45 | 0.30 | 0.38 | 0.84 | **0.92** |

Note: Bold numbers are the square roots of average variance extracted values.

coefficients with other latent variables (Table 2); this suggests that the factors' discriminant validity is good.

## Multicollinearity and autocorrelation tests

The F test finds F = 127.05 and the p-value = 0.000, which indicates the model passes the F test; its R square value is 0.88. This means that perceived risk, risk aversion, government initiatives, media influence, and destination image can account for 88% of changes in travel intention. As the multicollinearity test finds each variable's variance inflation factor (VIF) to be less than 5, it means there is no issue of multicollinearity. Meanwhile, the Durban-Watson (DW) statistic was 1.93, indicating that there is no autocorrelation and the model is well-constructed.

## Model fitting analysis

The arithmetic mean of the six perceived risk dimensions (physical, equipment, cost, social, performance, and psychological) serves as the observed variable indicating "perceived risk", and the structural equation model (SEM) is used to study the influence relationship between perceived risk, risk aversion, government initiatives, media influence, destination image, and travel intention. In the initial model, GFI and RMR are found to not meet the standards for optimal fit and in need of correction. According to the modification index (MI) calculated by AMOS, the MI of "government initiatives" affecting "media influence" is 100.613, while that of "media influence" affecting "government initiatives" is 100.633; this indicates a rather strong relationship between these two items. Some scholars have argued that such a covariance relationship between the government and the media. Establishing the relationship between these can raise the model's chi-square value, which is theoretically true, thus increasing correlation between the above mentioned variables.

The revised fitness index exhibits obvious improvements and greater adaptation. Its CMIN/DF is 2.33, GFI is 0.86, RMSEA is 0.06, RMR is 0.04, SRMR is 0.05, CFI is 0.95. These fitting indices—apart from GFI, which is close to 0.9—comprehensively demonstrate the model's favorable fit.

## Main effect analysis

Fig 4 shows the structural equation model. Destination image has a significant positive impact on travel intention ($\beta = 0.88$, p<0.001), so hypothesis 1 holds; perceived risk has a significant

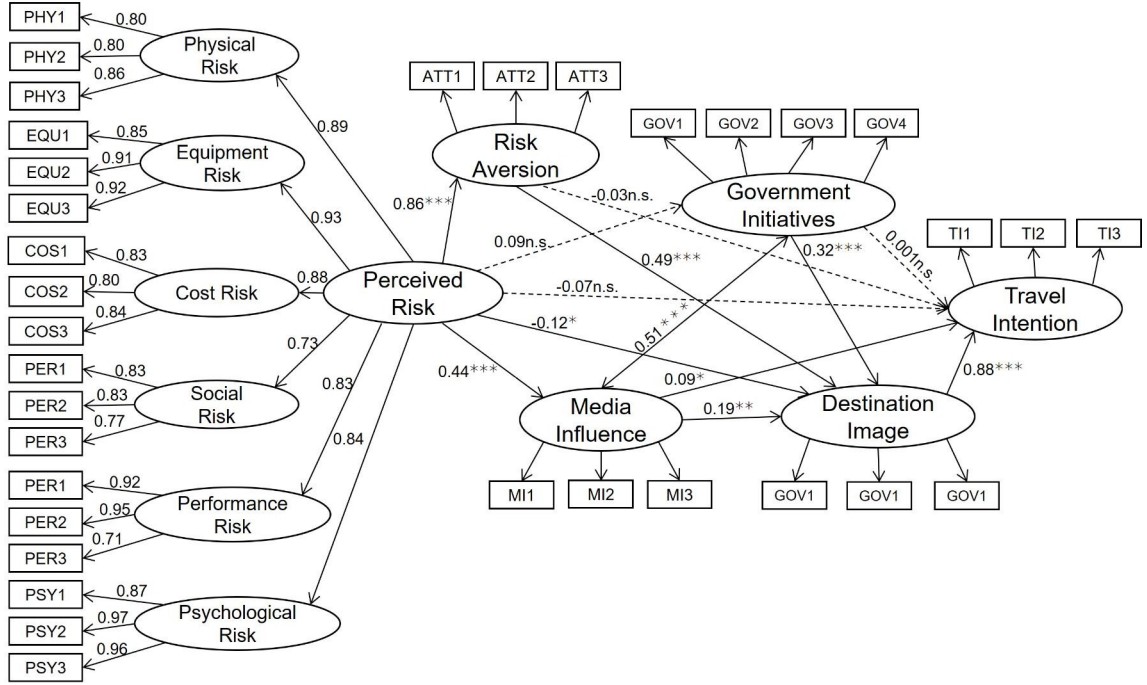

**Fig 4. Standardized output of the final integrated model.** Note: * p<0.05, ** p<0.01, *** p<0.001.

negative impact on destination image (β = −0.12, p<0.05), so hypothesis 2 holds; perceived risk has a negative but non-significant impact on travel intention (β = −0.07, p>0.05), so hypothesis 3 does not hold.

Path analysis is used to investigate different paths of influence (corresponding to various dimensions of perceived risk) on destination image and travel intention. All of the fit indices met the criteria with CMIN/DF = 2.5, GFI = 0.9, RMSEA = 0.06, RMR = 0.04, CFI = 0.94, NFI = 0.91, NFI = 0.91, NNFI = 0.94, which indicates the model is considered to fit well. As shown in Fig 5, physical risk, equipment risk, cost risk, and performance risk have significant negative impacts on destination image, while social and psychological risk have no significant negative impact on destination image; thus H2a holds in part. Physical risk, equipment risk, cost risk, social risk and performance risk have significant negative impacts on travel intention, while psychological risk has no significant negative impact on travel intention; thus H3a holds in part.

## Analysis of multiple mediating effects

The Bootstrap method is used to sample and analyze mediating effects, with a sampling frequency of 5,000. Results show that while perceived risk has a significant direct negative effect on destination image (β = −0.13, p<0.01), its total negative effect is not significant (β = −0.02, p>0.05).

Table 3 displays the analysis results. Two paths—"perceived risk→ risk aversion→ destination image" and "perceived risk→ government initiatives→ destination image"—meet the criteria of suppressing effects. Perceived risk impacts destination image indirectly through risk aversion and government initiatives, for which the suppressing effects are 61.91% and 25.39%, respectively. Therefore, H4 and H6 hold. The mediating effect of media influence, meanwhile, is insignificant; H8 does not hold. It can be seen that there are two significant paths of

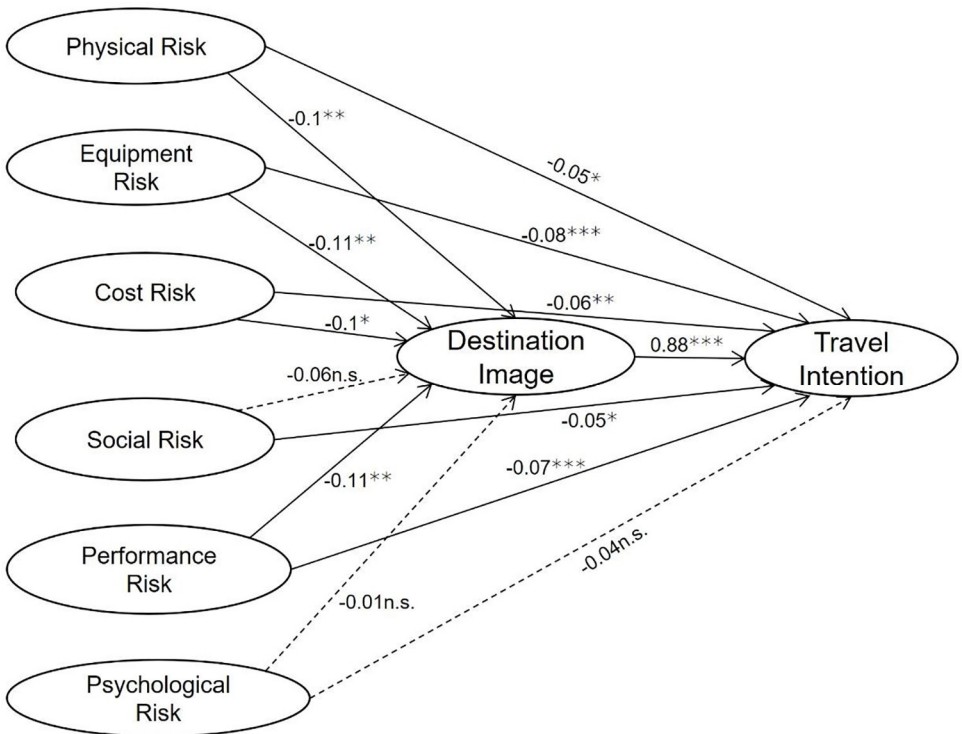

**Fig 5. Path analysis results.** Note: * p<0.05, ** p<0.01, *** p<0.001, n.s. p>0.05.

mediation by which perceived risk may affect destination image. Because the indirect effect (0.11) and direct effect (−0.13) are similar in magnitude but opposite in sign, the total effect is insignificant.

Among the various dimensions of perceived risk, physical risk, equipment risk, cost risk, and social risk affect destination image via government initiatives and risk aversion, effect

**Table 3. Analysis of indirect effects.**

| Item | c | a | b | a*b | a*b | c' | Conclusion | Effect Ratio |
|---|---|---|---|---|---|---|---|---|
| | Total Effect | | | Mediating Effect | (95% BootCI) | Direct Effect | | |
| Perc risk→Risk avers→Dest image | −0.02 | 0.20*** | 0.40*** | 0.08 | 0.01~0.14 | −0.13** | Suppressing effects | 61.91% |
| Perc risk→Govt init→Dest image | −0.02 | 0.13** | 0.25*** | 0.03 | −0.01~0.08 | −0.13** | Suppressing effects | 25.39% |
| Phys→Risk avers→Dest image | 0.01 | 0.22*** | 0.42*** | 0.09 | 0.05 ~ 0.17 | −0.10** | Suppressing effects | 89.89% |
| Phys→Govt init→Dest image | 0.01 | 0.09* | 0.25*** | 0.02 | −0.01 ~ 0.07 | −0.10** | Suppressing effects | 23.01% |
| Equipment→Risk avers →Dest image | −0.03 | 0.14** | 0.40*** | 0.06 | 0.01 ~ 0.13 | −0.11** | Suppressing effects | 54.11% |
| Equipment→Govt init →Dest image | −0.03 | 0.09* | 0.25*** | 0.02 | −0.01 ~ 0.07 | −0.11** | Suppressing effects | 21.58% |
| Cost→Risk avers →Dest image | 0.04 | 0.25*** | 0.41*** | 0.1 | 0.05 ~ 0.18 | −0.10* | Suppressing effects | 102.36% |
| Cost→Govt init →Dest image | 0.04 | 0.15*** | 0.26*** | 0.04 | 0.01 ~ 0.10 | −0.10* | Suppressing effects | 39.26% |
| Social→Risk avers →Dest image | 0.01 | 0.09* | 0.40*** | 0.04 | −0.01 ~ 0.10 | −0.06 | Full mediation | 100% |
| Social→Govt init→Dest image | 0.01 | 0.10** | 0.27*** | 0.03 | −0.00 ~ 0.08 | −0.06 | Full mediation | 100% |

* p<0.05

** p<0.01

*** p<0.001

Only the paths with mediating effects are shown. Insignificant mediation paths have been omitted.

**Table 4. Analysis of indirect effects.**

| Pathway | Effect | BootLLCI | BootULCI | Verdict |
|---|---|---|---|---|
| Perceived risk→ Risk aversion→ Travel intention | 0.00 | −0.009 | −0.001 | Y |
| Perceived risk→ Media influence→ Travel intention | 0.02 | 0.016 | 0.042 | Y |
| Perceived risk→ Risk aversion→ Govt initiatives→ Travel intention | 0.00 | 0.001 | 0.003 | Y |
| Perceived risk→ Risk aversion→ Media influence→ Travel intention | 0.00 | 0.001 | 0.006 | Y |
| Perceived risk→ Risk aversion→ Destination image→ Travel intention | 0.07 | 0.011 | 0.112 | Y |
| Perceived risk→ Risk aversion→ Government initiatives→ Media influence→ Travel intention | 0.00 | 0.001 | 0.008 | Y |
| Perceived risk→ Risk aversion→ Government initiatives→ Destination image→ Travel intention | 0.02 | 0.001 | 0.032 | Y |
| Physical risk →Media influence→ Travel intention | 0.01 | 0.010 | 0.030 | Y |
| Equipment risk→ Media influence→ Travel intention | 0.01 | 0.015 | 0.037 | Y |
| Social risk→ Media influence→ Travel intention | 0.01 | 0.008 | 0.027 | Y |
| Performance risk→ Media influence→ Travel intention | 0.01 | 0.012 | 0.032 | Y |
| Psychological risk→ Media influence→ Travel intention | 0.01 | 0.011 | 0.027 | Y |
| Cost risk→ Media influence→ Travel intention | 0.01 | 0.010 | 0.029 | Y |
| Cost risk→ Govt initiatives→ Travel intention | 0.01 | 0.001 | 0.007 | Y |
| Cost risk→ Government initiatives→ Media influence→ Travel intention | 0.01 | 0.002 | 0.015 | Y |
| Cost risk→ Government initiatives→ Destination image→ Travel intention | 0.05 | 0.011 | 0.105 | Y |

Note: "Y" means that there is significant mediation. Insignificant mediating effects are not shown.

ratio ranges from 21.58% to 102.36%. Therefore, H4a and H6a hold in part. The mediating effect of media influence is not significant for the various dimensions of perceived risk; therefore H8a does not hold.

The Bootstrap method is used to verify mediating effects of risk aversion, government initiatives, media influence, and destination image, with a sampling frequency of 5,000. The study finds that perceived risk has a significant direct negative impact on travel intention ($\beta = -0.09$, p<0.001), but that the total negative effect is not significant ($\beta = -0.07$, p>0.05).

The study uses chain mediation analysis to study the mediating effects of perceived risk on travel intention via risk aversion, government initiatives, media influence, and destination image. The results are shown in Table 4. Seven mediation paths are significant; among them, indirect effects (0.11) are greater than the direct effects (−0.09), leading to suppressing of the direct effects and net insignificance. Perceived risk has separate mediating effects on travel intention via risk aversion and media influence; therefore, H5 and H9 hold. Perceived risk has an insignificant effect, however, through government initiatives; H7 thus does not hold. As perceived risk has multiple mediating effects on travel intention via risk aversion, government initiatives, and destination image, H10 is found to hold.

Among the six dimensions of perceived risk, five of them—physical, equipment, social, performance, and psychological risk—have indirect impacts on travel intention via media influence. In addition to cost risk affecting travel intention through parallel mediation (media influence and government initiatives), it also affects travel intention through serial mediation (government initiatives→ media influence or destination image), H7a and H9a–H10a hold in part.

## Variance analysis

T-testing and ANOVA are used to investigate correspondence in the variation of perceived risk with differences in gender, marital status, age, education level, and place of origin. Results (Table 5) indicate that there is no significant difference in tourists' perceived risk according to

**Table 5. Difference analysis on varied demographic characteristics in perceived risk.**

| Variable | Group with Highest Perceived Risk | Group with Lowest Perceived Risk | Scheffe Post-hoc Test | Major Trend |
|---|---|---|---|---|
| | | | Significant Difference Factor | |
| Gender | Female | Male | female>male | Women perceive higher risk |
| Age | 41–50 | Under 18 | 18 and above>under 18; | Older people perceive higher risk |
| | | | 26–40>18–25 years old | |
| Education | Master's/Doctorate | Middle school or below | master's/doctorate>bachelor's/professional >high school/equivalent>middle school/below | More educated people perceive higher risk |
| Marital status | Married | Single | married>single | Married people perceive higher risk |
| Origin | Northeast China | Central China | none | – |

place of origin. Significant differences in perceived risk do exist, however, according to gender, marital status, age, and education level. Specifically: female>male; over 18>under 18 and 26–40>18–25; master's/doctoral degree>post-secondary professional/bachelor's degree>high school diploma>middle school or below; married>unmarried.

## Discussion

Generally speaking, perceived risk exerts a negative impact on intent to travel [10, 11]. However, perceived risk also influences travel intention indirectly through media influence and risk aversion, and more complexly through the serial mediation of government initiatives and destination image. As these indirect effects are opposite in sign to the direct effect of perceived risk, the direct effects are suppressed and the net effect becomes insignificant. Consistent with the findings of Tessitore et al [19], and Fu et al. [20], this study finds that destination image has a significant and positive impact on travel intention. Tourism destinations should therefore strive to establish favorable destination images. Meanwhile, this study finds that when impacted by a crisis, government initiatives have a greater effect on destination image than media influence (and media influence, it should be noted, has a positive effect). This diverges from Parrey's findings [68]—that in a conflict zone, media influence has a greater (and negative) effect on destination image than government initiatives. (While the current study, like Parrey's, verifies the negative impact of perceived risk on destination image and analyzes the mediation of the two abovementioned factors, it additionally verifies the negative impact of perceived risk on travel intention and adds "risk aversion" as another important mediating factor.)

Not all dimensions of perceived risk, however, exert a significant negative impact on destination image and travel intention [22, 43]. In this study, four dimensions—physical risk, equipment risk, cost risk, and performance risk—have a significant negative impact on both destination image and travel intention, while one dimension—social risk—has a significant negative impact on travel intention only. Psychological risk does not have a significant negative impact on either destination image or travel intention. This represents a slight departure from previous findings. For example, Chew et al. [22] find that physical risk does not impact destination image but that social and psychological risks do. The discrepancy may be due to differences in the research setting or cultural backgrounds of the participating human subjects. Chew et al.'s study is set against the background of the Fukushima nuclear disaster, and the main respondents are international tourists. This study, meanwhile, is set in the context of the COVID-19 pandemic, and survey respondents are domestic Chinese tourists.

Although physical, equipment, and cost risks have significant effects on destination image in this study, these effects are obscured by the interventions of risk aversion and government

initiatives. As for social risk, risk aversion and government initiatives completely mitigate its negative impact on destination image. Another mediating factor, media influence, counteracts the effect (on travel intention) of perceived physical risk, equipment risk, social risk, performance risk, and psychological risk, respectively. Cost risk not only affects travel intention through the actions of government initiatives and media influence individually; it also affects it through said actions' impact on destination image.

As in other studies [29–31, 82, 83], perceived risk is found to vary according to demographic variables and social statistics. Spence et al. [82] argue that a higher level of educational attainment corresponds to a lower perceived risk, which is due to the reduction of prejudices. In contrast, we conclude based on this study's evidence that the higher a tourist's education level, the better equipped they are to understand the effects of the novel coronavirus on both socioeconomic development and global tourism patterns; thus their perceived risk will be higher. Also, contrary to the conclusion of Roehl et al. [28], this study finds no significant difference in perceived risk among tourists of different source locations. This may be due to the nature of COVID-19 as a global public health event. As its impact truly covers the entire world, and people within a country share quite similar experiences and information, Chinese tourists' perceived risk does not differ significantly by their geographic origin.

## Conclusions

When it comes to exactly how perceived risk influences destination image and travel intention, the mechanism has long been unclear. This has caused insufficient understandings of crises' negative impacts, which in turn leads to neglect of the role that internal and external factors play in crisis management. Tourism crises are frequent occurrences, but the response in crisis management has been weak, seriously affecting tourism's ability to sustainably develop. Scholars have reached differing conclusions about the impact of perceived risk on behavioral intention. Most hold that perceived risk has a negative impact. A small number of empirical studies, however, present cases where perceived risk has insignificant negative or even a positive impact; still, such results suffer from a lack of reasonable explanation. Some scholars have speculated that these findings emerge because consumers actually focus more on maximizing utility than minimizing risk in their decision-making. We argue instead that "minimizing risk" is more complicated than previous studies are able to explain.

This study models the impact of perceived risk on destination image and travel intention, verifying the mediating roles of risk aversion, government initiatives, and media influence. Among these, government initiatives have a particularly significant positive impact during the COVID-19 crisis. Furthermore, this study breaks "perceived risk" down into six dimensions: physical risk, equipment risk, cost risk, performance risk, social risk, and psychological risk. The first five of the afore listed dimensions have a significant negative impact on travel intention, while the first four of them have a significant negative impact on destination image; however, risk aversion and government initiatives effectively negate the impact of social risk on destination image, while media influence neutralizes the negative impact of five dimensions (all save cost risk) on travel intention. Cost risk's impact on travel intention is mediated through risk aversion, government initiatives, and indirectly through destination image. Thus for Chengdu during the COVID-19 outbreak, the reason why tourism seemed relatively unaffected following a sudden outbreak was not due to the absence of influencing factors, but due to the complex interaction of influencing factors. In other words, it is only because of the government, media, and private citizens' deliberate risk response behavior that a reduced negative or even positive effect may be observed following a tourism crisis.

This study advances the literature on the mechanism of travel-related risk perception through a number of theoretical contributions. It provides an empirically verified conceptual framework that demonstrates travel behavior intentions in relation to risk perception, internal and external responses, and destination image. The literature on crisis and risk in tourism studies has failed to address a common mechanism for how risk perception is weakened by three intermediary variables—risk aversion, government initiatives, and media influence—and lead to positive destination images and decision-making. This study is significant because it provides a theoretical interpretation of tourism risk, elucidates the mechanisms or paths by which perceived risk affects travel intention, and expands a framework for research on destination image and travel intention into the realms of psychology, political, and communication science. It additionally encourages people to pay greater attention to the negative impact of crises and focuses on the important role of internal and external responses in crisis management, which can help improve the effectiveness of crisis management and promote the sustainable development of the tourism industry.

For a more detailed discussion, we divide perceived risk into six sub-risks, or dimensions: physical risk, equipment risk, cost risk, social risk, performance risk, and psychological risk. We then compare each dimension in terms of effect pathway, exploring perceived risk's impact on the boundary conditions for destination image and travel intention.

## Application

According to this study, perceived risk does have a negative impact on destination image and travel intention—but, due to the mediating and masking effects of risk aversion, government initiatives, and media influence, the overall effect of perceived risk is weakened. As all we can "see" is the net effect (in this case, the number of tourists after a crisis failing to drop significantly), the phenomenon is counterintuitive. This, however, is precisely why the tourism industry must devote attention to the mitigating role played by risk aversion, government initiatives, and media influence in tourism crisis management, and not underestimate the impact that tourists' cognitive or psychological factors have on a destination. Thus, in the post-crisis period, it is on the one hand necessary to make full use of the government's leading role in crisis management and brand-building—by inviting experts, scholars, or authoritative organizations to publicize the effectiveness of COVID-19 prevention and control and secure tourists' confidence in the destination, by diversifying tourism products and enriching tourism experiences, by formulating promotional policies and maintaining accurate marketing for key and potential markets, by extending special offers (as tourists' perception of cost risks are naturally the highest), as well as by strengthening supervision over the tourism industry, raising the level of performance, and setting up high-quality service brands. On the other hand, it is necessary to take full advantage of the positive promotional and informative role of the media. Official and mainstream media should concentrate on issuing positive reports in order to offset potential tourists' unfavorable perceptions following a crisis and thus change the image of the destination in question. They should continue to enhance tourism experiences, add value-added promotions, and provide tourists with timely and diverse travel information.

## Limitations of the study and directions for future research

This study models the impact of perceived risk on destination image and travel intention, taking the city of Chengdu during the COVID-19 pandemic as a case for empirical investigation. It must be noted, however, that crises and catastrophic events manifest in a variety of forms. According to the *Emergency Response Law of the People's Republic of China* [84], emergencies are divided into natural disasters, human-made disasters, public health incidents, and social

safety incidents. Our research can only represent public health incidents. Future research might analyze the impact of different types of emergency or of health incidents at different levels on destination image and travel intention, or compare specific differences with different emergencies. Meanwhile, this study focuses on perceived risk among domestic tourists in China. While future research might analyze perceived risk among international tourists and conduct a comparison. Finally, there are additional demographic or related variables that this study did not address—including family composition, age of dependents, distance from destination, frequency of past behavior, and visiting history—that may lead to differential tourist responses, and may be studied in-depth in future works.

## Supporting information

**S1 Table. Similar studies in China.**
(DOCX)

**S1 File.**
(DOCX)

## Acknowledgments

The authors would like to thanks the editors and the reviewers for their efforts and constructive feedback, which helped improve the manuscript. The authors would also like to appreciate the respondents for their kind participation in the research.

## Author Contributions

**Conceptualization:** Xiufang Jiang, Jianxiong Qin.

**Data curation:** Xiufang Jiang, Jianguo Gao.

**Formal analysis:** Xiufang Jiang, Jianguo Gao.

**Funding acquisition:** Xiufang Jiang, Jianxiong Qin.

**Investigation:** Xiufang Jiang, Jianguo Gao.

**Methodology:** Xiufang Jiang.

**Project administration:** Xiufang Jiang, Jianxiong Qin.

**Resources:** Xiufang Jiang.

**Software:** Xiufang Jiang, Jianguo Gao.

**Supervision:** Jianxiong Qin, Mollie G. Gossage.

**Validation:** Xiufang Jiang.

**Visualization:** Xiufang Jiang, Jianguo Gao.

**Writing – original draft:** Xiufang Jiang.

**Writing – review & editing:** Xiufang Jiang, Jianxiong Qin, Jianguo Gao, Mollie G. Gossage.

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
