## [Decision Letter · Decision Letter 0]

21 Oct 2021

PONE-D-21-28608The Mediation of Perceived Risk’s Impact on Destination Image and Travel Intention: An Empirical Study of Chengdu, China during COVID-19PLOS ONE

Dear Dr. Jiang,

Thank you for submitting your manuscript to PLOS ONE. After careful consideration, we feel that it has merit but does not fully meet PLOS ONE’s publication criteria as it currently stands. Therefore, we invite you to submit a revised version of the manuscript that addresses the points raised during the review process.

We look forward to receiving your revised manuscript.

Kind regards,

Ghaffar Ali, PhD

Academic Editor

PLOS ONE

Journal Requirements:

Reviewers' comments:

Reviewer's Responses to Questions

**Comments to the Author**

1. Is the manuscript technically sound, and do the data support the conclusions?

Reviewer #1: Yes

Reviewer #2: Yes

2. Has the statistical analysis been performed appropriately and rigorously? 

Reviewer #1: Yes

Reviewer #2: Yes

3. Have the authors made all data underlying the findings in their manuscript fully available?

Reviewer #1: Yes

Reviewer #2: Yes

4. Is the manuscript presented in an intelligible fashion and written in standard English?

Reviewer #1: Yes

Reviewer #2: Yes

5. Review Comments to the Author

Reviewer #1: I thoroughly enjoyed reading the manuscript and really appreciate the detailed literature review to build and support the hypotheses and excellent theoretical and analytical approaches and the discussion of the results. While I have no doubt about the merit of the research and its methodological rigor, as a reader from outside PR China, I would have liked to know bit more about study site through location map and general variation in local tourist numbers in different months before and after COVID. Similarly the spread of respondents on a map would help to understand geographical location of respondents and distance from the study site. Similar some more description about the 'Questionnaire Star' and/or a weblink for the readers to know its recommendation process etc. Regarding demographic and related variables, I would like to mention that family composition and age of dependent kids or other family members could have provided useful information - though part of issue might have been addressed in social risks SOC2. The respondents have not been asked about frequency of their travel in before corona period as that could have explained how an otherwise frequent traveler responds. Similarly those have already visited the study site might have a different perception about intent to travel and destination image etc. Similarly it has not been captured that those form nearby areas who have been frequently visiting the have more or less urge than distant respondents and a variation in their responses. The questionnaire is well elaborated, however some instrument items like in Social Risk factor (SOC1, SOC2, SOC3) or Equipment Risk (EQ1, EQ2, EQ3) seems confusing and there may be high likelihood that the respondent might have been confused in responding some items (For example if other think negative Vs if friends and family members thinks negative in Social Risks; in Equipment Risk for example it might be confusing to differentiate between poor basic infrastructure followed by perception of poor sanitation and traffic inconvenience. I believe the questionnaire was in Chinese language and the translation in English could have created some confusion. Despite these minor comments, the study is an excellent effort to capture COVID-19 related barriers to tourism through mediated and direct impacts.

Reviewer #2: Overall a good approach and analysis presented. I would like to provide with following comments and suggestions quickly:

1. Abstract should be revised and add more results and practical policy implications.

2. Try to reduce some information in the first section, it is very lengthy.

3. In methodology, what is reliability of data?

4. How biasness was addressed and set?

5. why this sample number was selected and how? which method used and why?

6. Discussion heading and after that "findings", inappropriate. Discussion should not contain findings. only discuss and compare with other such similar studies.

7. For conclusions also follow my comments as for abstract.

8. Add table on the comparison of such studies in Sichuan or nearby provinces in China and if not then from other cities of China.

9. Also better to give a map of Chengdu and show there data collection points and sampling.

10. Add more figures if possible for better understanding and presentation.

6. PLOS authors have the option to publish the peer review history of their article (what does this mean?). If published, this will include your full peer review and any attached files.

Reviewer #1: **Yes: **Muhammad Asif Kamran

Reviewer #2: No

---

## [Author Response · Author response to Decision Letter 0]

3 Nov 2021

Thank you for your valuable comments concerning our manuscript entitled “The Mediation of Perceived Risk’s Impact on Destination Image and Travel Intention: An Empirical Study of Chengdu, China during COVID-19”. These comments are all very helpful for the revision and improvement of our paper, as well as regards the guiding significance of our research. We have studied the comments carefully and made corrections which we hope will meet with approval. Our responses to these comments are shown in "Response to Reviewers" file.

---

## [Decision Letter · Decision Letter 1]

13 Dec 2021

The mediation of perceived risk’s impact on destination image and travel intention: An empirical study of Chengdu, China during COVID-19

PONE-D-21-28608R1

Dear Dr. Jiang,

We’re pleased to inform you that your manuscript has been judged scientifically suitable for publication and will be formally accepted for publication once it meets all outstanding technical requirements.

Kind regards,

Ghaffar Ali, PhD

Academic Editor

PLOS ONE

Additional Editor Comments (optional):

Reviewers' comments:

Reviewer's Responses to Questions

**Comments to the Author**

1. If the authors have adequately addressed your comments raised in a previous round of review and you feel that this manuscript is now acceptable for publication, you may indicate that here to bypass the “Comments to the Author” section, enter your conflict of interest statement in the “Confidential to Editor” section, and submit your "Accept" recommendation.

Reviewer #1: All comments have been addressed

Reviewer #2: All comments have been addressed

2. Is the manuscript technically sound, and do the data support the conclusions?

Reviewer #1: Yes

Reviewer #2: Yes

3. Has the statistical analysis been performed appropriately and rigorously? 

Reviewer #1: Yes

Reviewer #2: Yes

4. Have the authors made all data underlying the findings in their manuscript fully available?

Reviewer #1: (No Response)

Reviewer #2: Yes

5. Is the manuscript presented in an intelligible fashion and written in standard English?

Reviewer #1: Yes

Reviewer #2: Yes

6. Review Comments to the Author

Reviewer #1: (No Response)

Reviewer #2: The changes have been made according to the comments and suggestions provided earlier. I am happy with the revised version, therefore, I allow for publication.

7. PLOS authors have the option to publish the peer review history of their article (what does this mean?). If published, this will include your full peer review and any attached files.

Reviewer #1: **Yes: **Muhammad Asif Kamran

Reviewer #2: No

---

## [Editor Report · Acceptance letter]

21 Dec 2021

PONE-D-21-28608R1 

The mediation of perceived risk’s impact on destination image and travel intention: An empirical study of Chengdu, China during COVID-19 

Dear Dr. Jiang:

I'm pleased to inform you that your manuscript has been deemed suitable for publication in PLOS ONE. Congratulations! Your manuscript is now with our production department. 

Kind regards, 

on behalf of

Prof. Ghaffar Ali 

Academic Editor

PLOS ONE